# Developing a Deep Learning-Based Defect Detection System for Ski Goggles Lenses

**Dinh-Thuan Dang [1,2] and Jing-Wein Wang [3,\*]**

[1] Department of Electronics Engineering, National Kaohsiung University of Science and Technology, Kaohsiung 80778, Taiwan

[2] Department of Information Technology, Pham Van Dong University, Quang Ngai 57000, Vietnam

[3] Institute of Photonics Engineering, National Kaohsiung University of Science and Technology, Kaohsiung 80778, Taiwan

\* Correspondence: jwwang@nkust.edu.tw

**Abstract:** Ski goggles help protect the eyes and enhance eyesight. The most important part of ski goggles is their lenses. The quality of the lenses has leaped with technological advances, but there are still defects on their surface during manufacturing. This study develops a deep learning-based defect detection system for ski goggles lenses. The first step is to design the image acquisition model that combines cameras and light sources. This step aims to capture clear and high-resolution images on the entire surface of the lenses. Next, defect categories are identified, including scratches, watermarks, spotlight, stains, dust-line, and dust-spot. They are labeled to create the ski goggles lenses defect dataset. Finally, the defects are automatically detected by fine-tuning the mobile-friendly object detection model. The mentioned defect detection model is the MobileNetV3 backbone used in a feature pyramid network (FPN) along with the Faster-RCNN detector. The fine-tuning includes: replacing the default ResNet50 backbone with a combination of MobileNetV3 and FPN; adjusting the hyper-parameter of the region proposal network (RPN) to suit the tiny defects; and reducing the number of the output channel in FPN to increase computational performance. Our experiments demonstrate the effectiveness of defect detection; additionally, the inference speed is fast. The defect detection accuracy achieves a mean average precision (mAP) of 55%. The work automatically integrates all steps, from capturing images to defect detection. Furthermore, the lens defect dataset is publicly available to the research community on GitHub. The repository address can be found in the Data Availability Statement section.

**Keywords:** ski goggles lenses; surface defect; automatic optical inspection; Faster-RCNN; fine-tune; MobileNetV3; FPN; RPN

**MSC:** 68T07, 68T20, 68T45

## 1. Introduction

Winter sports such as skiing, snowboarding, and snowshoeing offer great enjoyment, and ski goggles are the necessary equipment to perform better in these activities. There are many advantages that can help protect the eyes from harmful ultraviolet rays, provide both facial and ocular safety protection, and offer color and contrast enhancement. Figure 1 shows some samples of ski goggles.

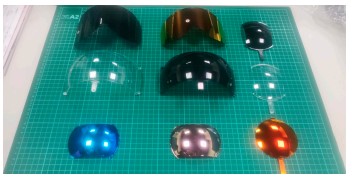

**Figure 1.** Some lens samples of ski goggles in different sizes, curvatures, and colors.

The most critical component of ski goggles is the lenses, which offer an unrivaled visual experience. In the manufacturing process of ski goggles, there are unavoidable defects [1,2] on the surface of the lenses. Therefore, lens manufacturers should implement a visual defect inspection system [3,4] to enhance product quality.

There are some challenging issues in the defect inspection for the lenses of ski goggles. Firstly, the lens surfaces are curved and vary in size, making it difficult to design an image acquisition system that captures their entire surface. As depicted in Figure 2, the usual lens samples exhibit distinct curvatures and sizes. Secondly, the lenses are coated with various tints, presenting a challenge in customizing the light source due to multiple reflections. Thirdly, some surface defects are extremely small, which are inconspicuous and subtle. Thus, the detection of such minor defects is particularly difficult and requires higher precision.

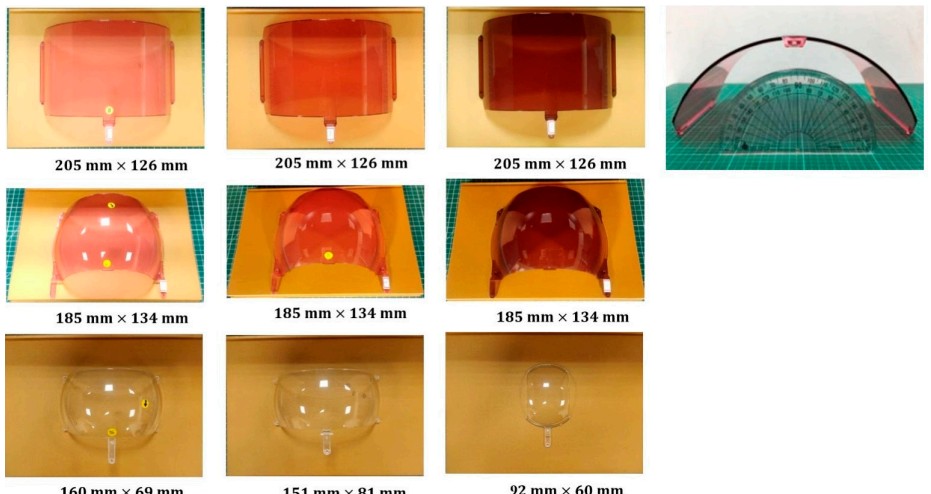

**Figure 2.** Some standard sizes of ski goggles lenses. The width ranges from 92 to 205 mm, and the height ranges from 60 to 126 mm.

Artificial intelligence (AI) technology can help organizations gain an edge over their competitors [5]. AI has proven especially beneficial for improving product quality and lowering costs. For manufacturers, AI promises benefits at every level of the value chain. The AI-based system can detect defects faster and more accurately than the human eye. A typical industrial visual inspection system based on deep learning incorporates several fundamental components to facilitate accurate and efficient product quality assessment. These components, critical to the system's operation, include:

- Image Acquisition Devices: High-resolution cameras [6,7], often fitted with specialized lenses, capture images of inspected items. These devices may employ various imaging technologies, such as monochrome, color, or infrared, contingent on the application's demands.
- Lighting: Customized illumination sources [8], including LED lights [9] or lasers, are employed to enhance the contrast and visibility of features under inspection. The choice and configuration of lighting are instrumental in achieving optimal image quality for precise defect detection.
- Deep Learning Algorithms: Deep learning-based defect detection typically requires training object detection models or alternative specialized architectures on the extensively labeled datasets of defect images. Object detection methodologies have been extensively applied in the detection of defects on the surfaces of industrial products, such as steel, plastic, wood, and silk [10–12]. The task of object detection in computer vision encompasses two primary functions: localization [13] and classification [14]. In traditional computer vision, classifiers [15] such as SVM, KNN, and K-means clus-

tering have played a vital role in categorizing classes. Meanwhile, object localization mainly employs fast template-matching-based algorithms [16].

In recent years, significant progress has been made in neural networks, machine learning, and deep learning. Classifiers such as ResNet [17], VGG [18], EfficientNet [19], and Vision Transformer [20] have achieved state-of-the-art results for classification tasks. Concurrently, object localization has been applied to anchor-based and anchor-free methods [21]. The anchor-based approach, known as the two-stage object detector, includes the Faster R-CNN [22] family. The anchor-free technique, referred to as the one-stage object detectors, comprises models such as YOLO [23] and FCOS [24].

The remarkable success in this field stems from the seamless integration of localization and classification tasks in deep learning. To gain a deeper insight, it is crucial to understand the Faster R-CNN architecture's automatic pipeline. The Faster R-CNN model systematically combines customizable neural sub-network blocks, including the backbone block, region proposal network [25], and ROI-head block.

Figure 3 illustrates the fundamental components of an optical initialization system, which include a light source, camera, and hardware to execute image processing algorithms.

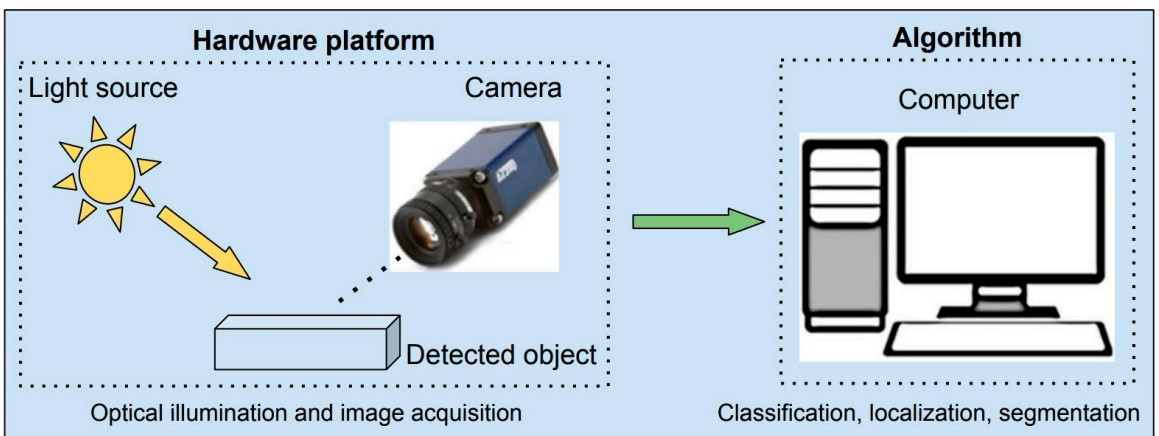

**Figure 3.** Some main components in the optical inspection system.

In light of the aforementioned challenges and emerging trends, this paper aims to develop an automatic defect detection system for ski goggles lenses, utilizing deep learning techniques. To accomplish this objective, the following steps are undertaken:

(1) Design of an image acquisition model that integrates cameras and light sources to effectively capture the entire surface of ski goggles lenses.

(2) Identification of lens defect categories and construction of a comprehensive ski goggles lens defect dataset.

(3) Fine-tuning of the integrated object detection model that combines Faster R-CNN, FPN, and MobileNetV3 by implementing the following modifications: replacement of the default ResNet50 backbone with a combination of MobileNetV3 and feature pyramid network (FPN) to optimize computational efficiency and performance; adjustment of the region proposal network (RPN) hyperparameters to accommodate the detection of minuscule defects; and a reduction of the output channel count in the FPN to enhance computational performance without sacrificing accuracy.

By executing these steps, the paper presents a novel deep learning-based approach for detecting defects on ski goggles lenses, demonstrating potential applicability to various manufacturing quality control scenarios.

The structure of the paper is organized as follows. Section 1 provides an introduction to the study. Section 2 introduces the image acquisition technique, data labeling for defects, and the defect detection method. Sections 3 and 4 present the research results

and subsequent discussions, respectively. Finally, Section 5 offers concluding remarks and summarizes the paper's findings.

## 2. Materials and Methods

The deep learning-based defect detection system for ski goggles lenses presented in this study is depicted in Figure 4. The system comprises two modules: the image acquisition module and the defect detection module. The former is responsible for capturing images of ski goggles lenses, extracting regions of interest, and labeling data. The latter trains the customized model using input data and detects defects during inference.

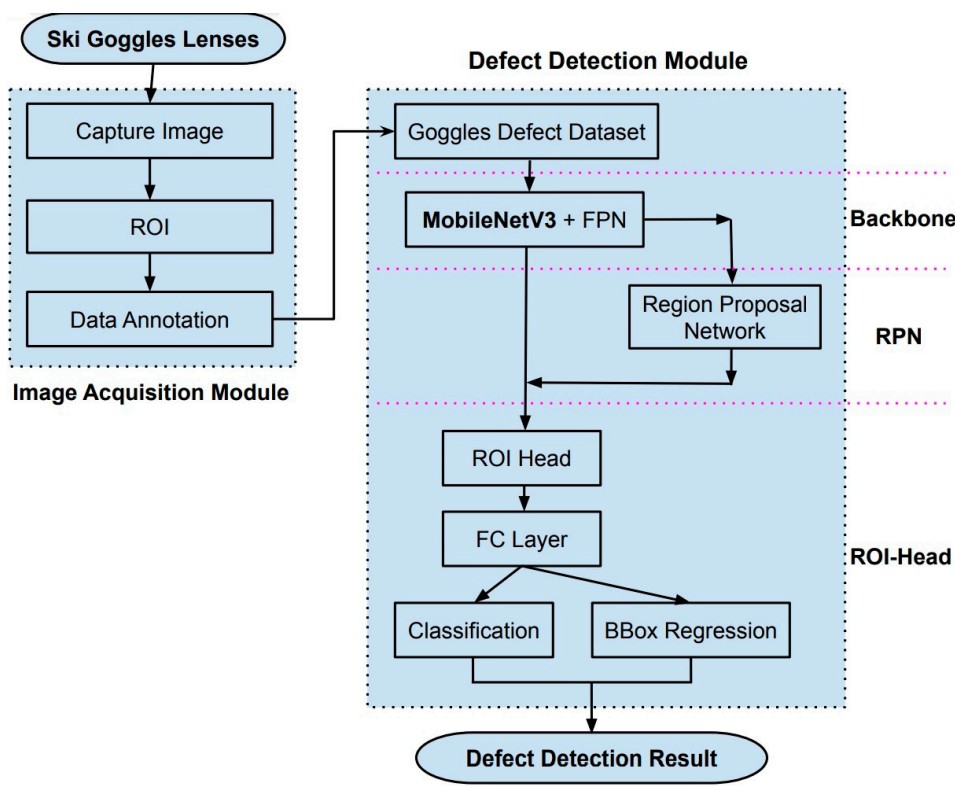

**Figure 4.** The flowchart of the proposed method. There are two modules for processing images. The first one is the image acquisition to capture the raw image, extract regions of interest, and data labeling. The second module is defect detection, which involves training data and inferencing defects. This module combines Faster R-CNN, MobileNetV3, and FPN to create the customized end-to-end model. It is compatible with data of the ski goggles lenses.

Image Acquisition Module: This module is responsible for obtaining high-quality images of ski goggles lenses. An optimized image acquisition setup, which combines cameras and light sources, is employed to ensure the entire surface of the lenses is captured with minimal glare and distortion. The regions of interest are extracted from the captured images, and the data are meticulously labeled to identify and categorize defects present in the lenses.

Defect Detection Module: A customized object detection model, based on the Faster R-CNN architecture, is designed and integrated into this module. The model involves replacing the backbone of Faster R-CNN with MobileNetV3 and integrating FPN for efficient feature extraction and multi-scale representation. The customized model is trained using the labeled input data and subsequently employed for detecting defects during the inference phase.

### 2.1. Image Acquisition Module

This module aims to collect accurate and high-quality images from the surface of the ski goggles lenses. It also prepares the well-formatted data for the next module.

### 2.1.1. Capture Image

The first part of the module is image capture, which consists of cameras, light sources, and ski goggles lens samples. To design the image capture system for the ski goggles, lenses need to overcome some challenges mentioned in Section 1. The surface of the lens is broad, and one camera cannot cover the whole of the lens surface. Therefore, we designed the image acquisition system using five cameras. Each camera will focus on each region marked on the lens, as in Figure 5.

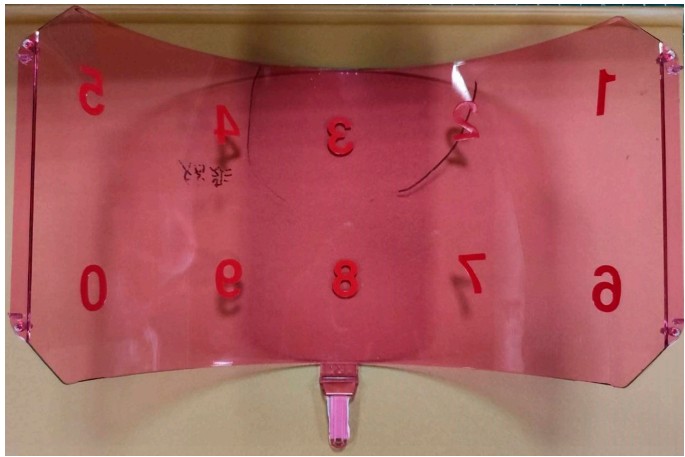

**Figure 5.** The ski goggles' lens sample. It is wide and curved; thus, we mark its surface to be easily controlled by cameras.

Furthermore, the surface is also curved, so we designed the custom light source as in Figure 6. The curvature of the light source is similar to the curvature of the lens, which helps to reflect uniform rays over the lens surface. The custom light source has five pieces of flat LED lights connected by an angle of 125 degrees. Figure 7 describes the detailed design diagram of the image acquisition system. The ray of each flat LED piece transits through the lens to the cameras opposite, respectively. Figure 8 depicts the actual pieces of equipment when deployed.

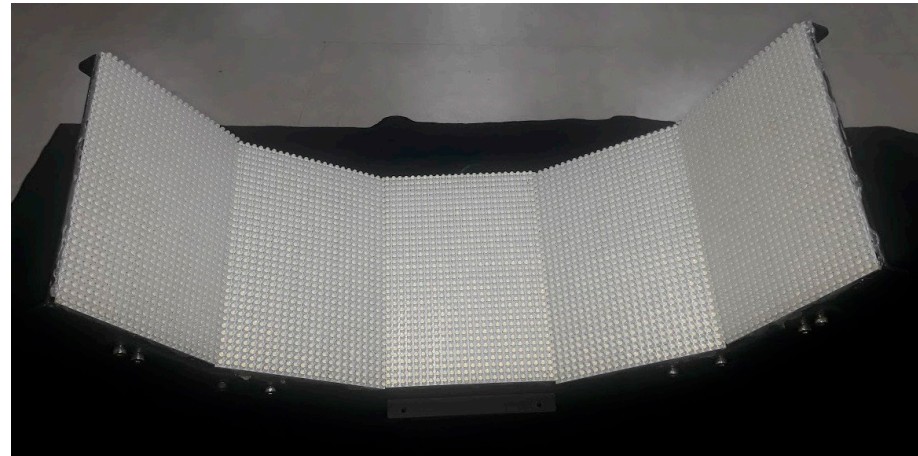

**Figure 6.** The most that a custom light source meets the curvature of the ski goggles lens. Five dot matrix LED modules are connected by an angle of 125°.

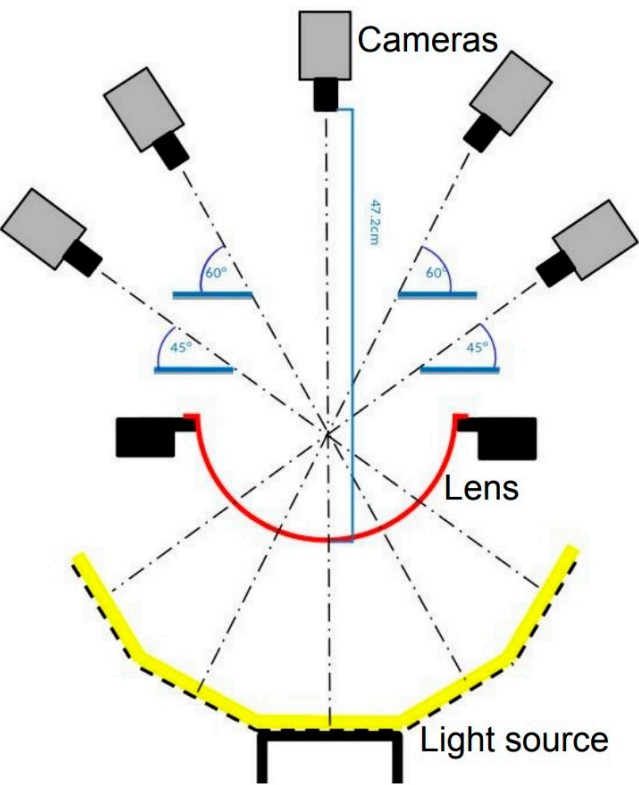

**Figure 7.** Design diagram of the image acquisition system. Five cameras are placed at the top. The custom light source (yellow) is placed at the bottom. The ski goggles' lens sample (red) is positioned in the middle and held on two sides.

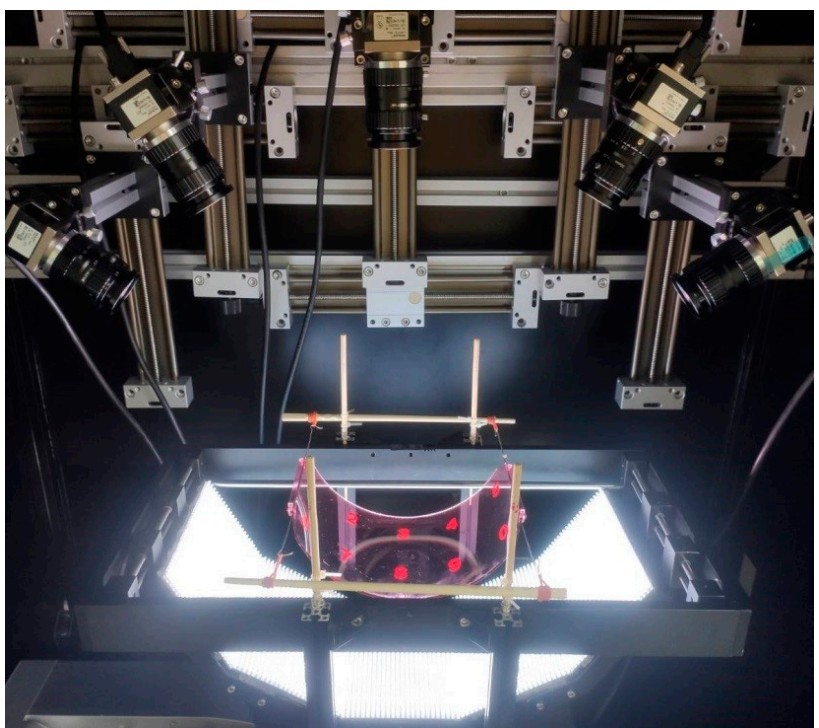

**Figure 8.** The actual model of image acquisition. The bottom is a custom light source that shines through the ski goggles lens surface to the cameras. The computer controls the five cameras through the acquisition card. The developed program will access the card's interface to capture images simultaneously.

The parameters of a camera, such as resolution, sensor, pixel size, and frame rate, are significantly influential in designing the distance from the camera lens to the inspected object. Table 1 lists the parameters of the image acquisition system.

**Table 1.** The equipment description of the image acquisition system.

| Equipment | Producer | Specification |
|---|---|---|
| Camera | Basler | Model acA4112-30uc, sensor Sony IMX352, resolution 12 mp, pixel size 3.45 × 3.45 μm, frame rate 30 fps. |
| Acquisition Card | Basler | USB 3.0 Interface Card PCIe, Fresco FL1100, 4HC, x4, 4Ports. Data transfer with rates of up to 380 MB/s per port. |
| Vision Lens | Tokina | Model TC3520-12MP, image format 4/3 inch, mount C, focal length 35 mm, aperture range F2.0-22. |
| Light Source | Custom | The custom-designed light source comprises five-dot matrix LED modules that are connected by an angle of 125°. |
| Computer | Asus | Windows 10 Pro; hardware based on: mainboard Asus Z590-A, CPU Intel I7-11700K, RAM 16G, VGA gigabyte RTX 3080Ti 12 GB. |

For ease of visualization, Figure 9 shows the input and output of the system. Inputs are lens samples. The system captures its surface and outputs images of the lens surface. We also developed a Python program to simultaneously control five cameras and automatically crop the areas of interest (ROI).

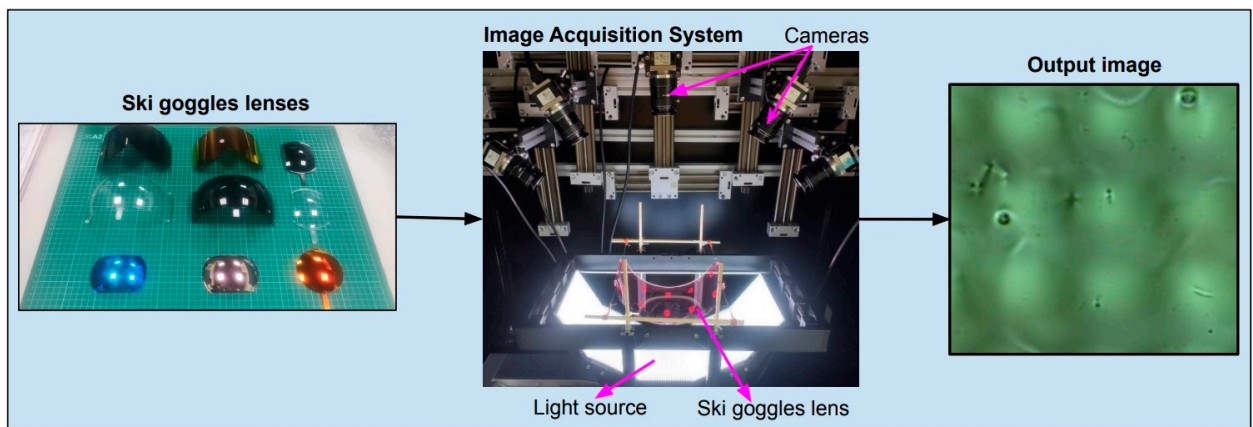

**Figure 9.** Input and Output of image acquisition. Input is some lens samples of ski goggles, and output is raw images from five cameras.

### 2.1.2. Regions of Interest

The system uses five cameras to capture the whole of the lens surface, and each camera only focuses on a portion of the lens. There are, however, limitations in the experiment, such as that the cameras can capture the overlapping or out-boundary parts. Therefore, we need to generate the ROIs from each raw image so that when stitched together, they become the image of the whole lens. The first line of Figure 10 shows five natural photos taken from the cameras, each containing redundant portions such as overlaps or areas outside of the lens. The five below images are the results of creating ROIs, respectively. We developed a program to capture images and generate ROIs seamlessly. The program was inherited from the Pypylon package of Basler and PyTorch framework.

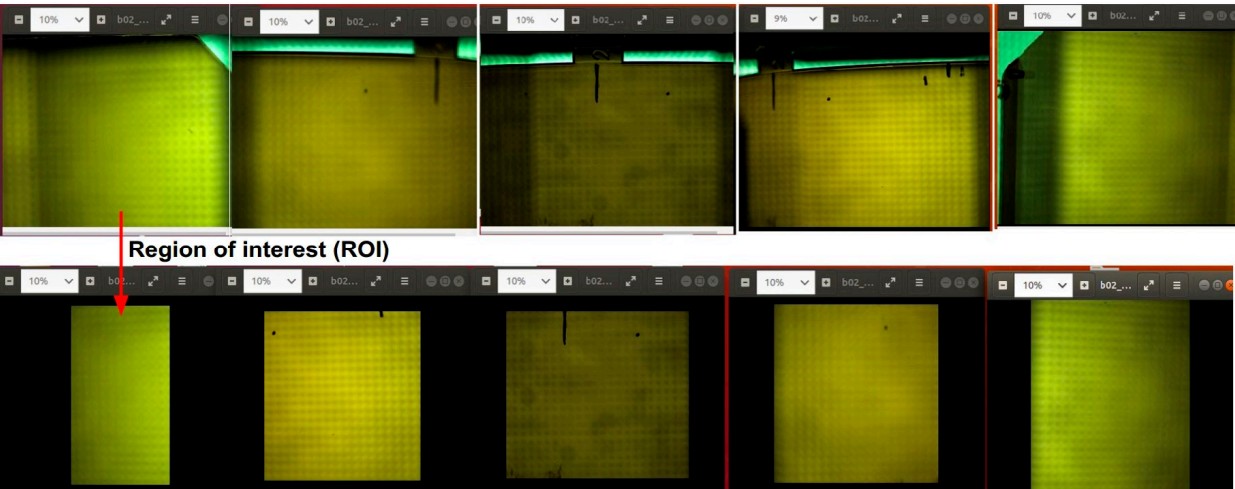

**Figure 10.** Crop regions of interest from raw images. Five cameras capture five images in the first row. To facilitate data labeling, extracting parts of good images is necessary. The **bottom** row is five regions of interest.

### 2.1.3. Data Labeling

Based on our practical experience with the imaging system and discussing with the ski goggles manufacturer, we conclude that there are the following common types of defects on the surface of ski goggles lenses: scratch, watermark, spotlight, stain, dust-line, and dust-spot. Figure 11 illustrates the detailed defect types.

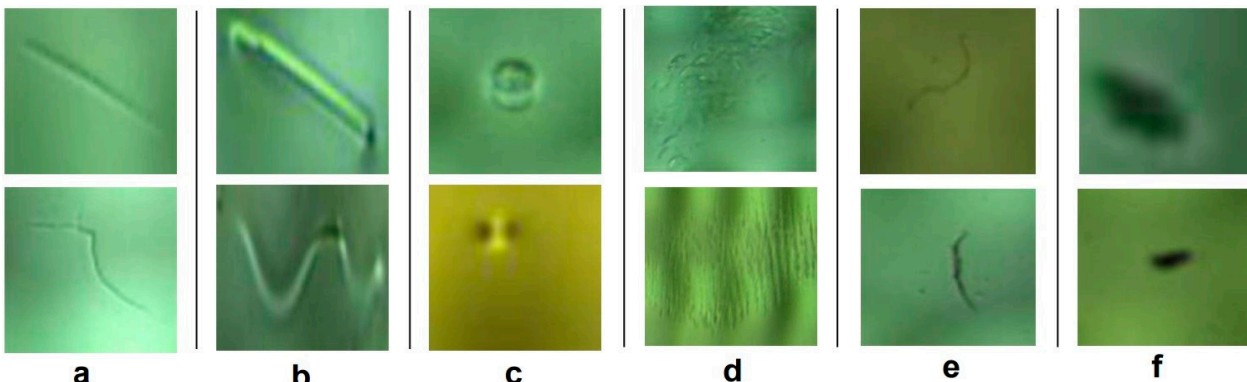

**Figure 11.** Some defect samples on the surface of ski goggles lenses. Column (**a**): scratch defects, (**b**): watermark, (**c**): spotlight, (**d**): stain, (**e**): dust-line, (**f**): dust-spot.

Because detect detection is based on supervised deep learning methods, the image data need to be labeled for the training phase. We used the LabelMe tool [26] to mark the defect regions with bounding boxes. Figure 12 illustrates the defect-labeling interface using the label tool.

From the 37 ski goggles lens pieces provided by the manufacturer, the image acquisition system captured and created a total of 654 images of $1330 \times 800$ pixels in size. We carry out defect labeling for the defect detection task. As outlined in Table 2, the count of labeled defects constitutes the initial dataset.

It is crucial to acknowledge that the distribution of defects in the dataset is imbalanced, with dust-line being the most prevalent (7292 instances) and watermark being the least prevalent (120 instances). This imbalance may result in a biased model. Consequently, the flip technique is employed to generate supplementary synthetic data. The quantity of underrepresented defect categories, including spotlight, stain, and watermark, is expected to increase. As depicted in Figure 13, the synthetic image is generated from a small

batch of defects and backgrounds. A total of 200 synthetic images are generated and subsequently labeled. Table 3 presents the statistics regarding the categories of defects in the synthetic dataset.

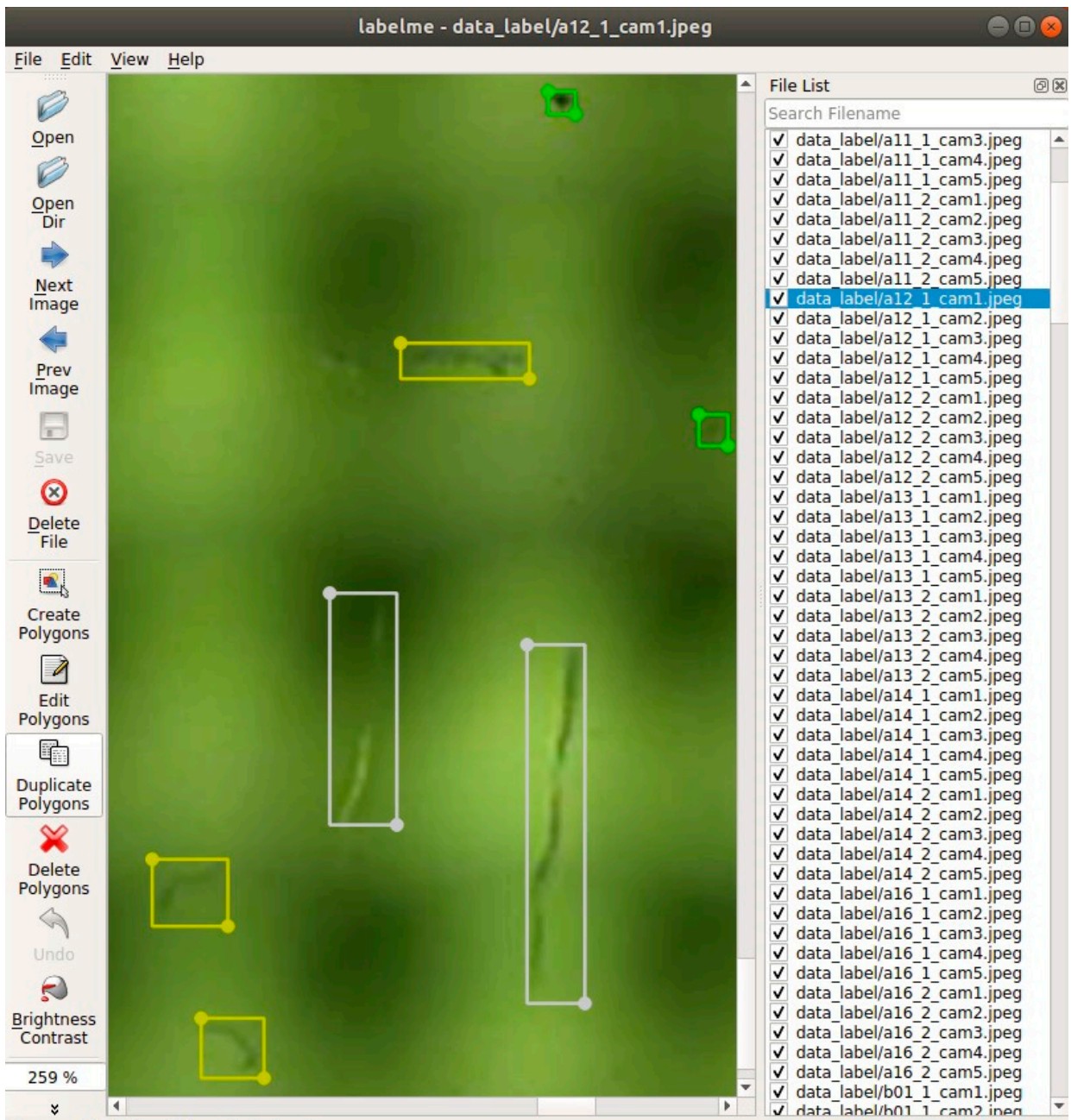

**Figure 12.** The GUI of LabelMe: the image annotation tool used to label defects on the surface of ski goggles lenses.

**Table 2.** The defect detection dataset of ski goggles lenses: defect type and its respective quantity.

| Type | Defects | Type | Defects | Type | Defects |
|------|---------|------|---------|------|---------|
| scratch | 1972 | spotlight | 229 | dust-line | 7292 |
| watermark | 120 | stain | 281 | dust-spot | 1898 |
| Total | 11,792 | | | | |

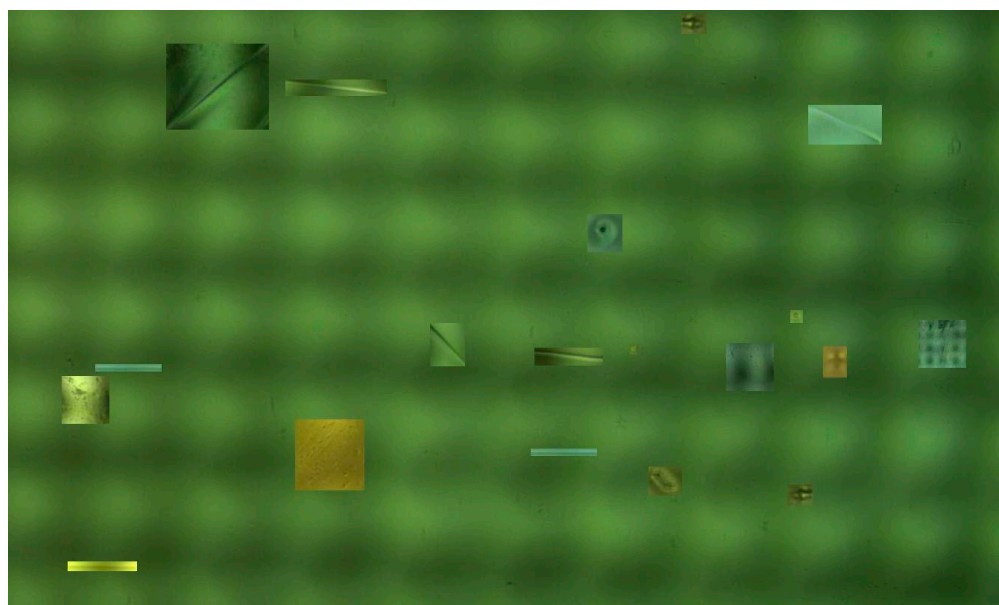

**Figure 13.** Synthetic image generated from the flip technique.

**Table 3.** The statistics of the number of defects in the synthetic dataset.

| Type | Defects | Type | Defects | Type | Defects |
|------|---------|------|---------|------|---------|
| scratch | 0 | spotlight | 1093 | dust-line | 0 |
| watermark | 973 | stain | 1328 | dust-spot | 0 |
| Total | 3394 | | | | |

Imbalanced and underrepresented data are a common challenge in collecting real-world data, which may impact detection results. To address this issue, a synthetic dataset is generated and subsequently merged with the initial dataset, forming a combined dataset as detailed in Table 4.

**Table 4.** The statistics of the defect categories from the combined dataset, which merges the initial and synthetic datasets.

| Type | Defects | Type | Defects | Type | Defects |
|------|---------|------|---------|------|---------|
| scratch | 1972 | spotlight | 1322 | dust-line | 7292 |
| watermark | 1093 | stain | 1609 | dust-spot | 1898 |
| Total | 3394 | | | | |

For a comprehensive and in-depth analysis of the dataset, Table 5 enumerates the number of images corresponding to each defect category.

**Table 5.** The number of images containing each defect category is extracted from the JSON file containing the labels.

| Defect Type | Scratch | Watermark | Spotlight | Stain | Dust-Line | Dust-Spot |
|-------------|---------|-----------|-----------|-------|-----------|-----------|
| Images | 447 | 199 | 352 | 316 | 546 | 612 |
| Instances | 1972 | 1093 | 1322 | 1609 | 7292 | 1898 |

The next section will describe the defect detection model, which is trained and utilized for inference using the aforementioned dataset.

### 2.2. Defect Detection Module

Finding a suitable object detection model for each data type is difficult. Faster R-CNN architecture is the two-state object detector which has proven to have high accuracy and be end-to-end trainable. Our work is to fine-tune this architecture by integrating the MobileNetV3 [27] backbone and feature pyramid network for extracting multi-scale features [28]. MobileNetv3 model is a lightweight neural network suitable for devices with a limited computational resource budget. Furthermore, we also reduce the number of channels to reduce latency in inference.

The following sections will cover the overview method of supervised machine learning theory, and the overview of the integrated Faster-RCNN architecture is shown in Figure 4. The backbone, RPN, and ROI-Head are the three main sub-networks in the defect detection module. First, the backbone block combines MobilenetV3 and FPN to extract multi-scale feature maps. Second, the RPN will create and propose the candidate defect regions. Finally, the ROI-Head block will locate the position of defects and classify them. Related theories, such as the bounding box regression, binary classification, multiclass classification, and assigning the boxes to the level of feature maps, are also discussed in detail.

#### 2.2.1. Object Detection Problem Setting Based on Supervised Learning Approach

There are various machine-learning paradigms, such as supervised, unsupervised, and reinforcement learning. Because of the tasks related to detecting and classifying defects, we apply the supervised learning approach. This direction is related to the input data, labels, generative networks, loss functions, and measure metrics. This section describes the basic theory of supervised learning.

Description: When given an image, determine whether or not there are instances of objects from predefined classes and, if present, return the bounding box of each instance.

Input: A collection of $N$ annotated images $X_{train}$ and a label set $Y_{train}$.

$$X_{train} = \{x_1, x_2, \ldots, x_N\} \tag{1}$$

$$Y_{train} = \{y_1, y_2, \ldots, y_N\} \tag{2}$$

where $y_i$ is annotation in image $x_i$, and each $y_i$ has $M_i$ objects belong to $C$ classes.

$$y_i = \left\{ (b_1^i, c_1^i), (b_2^i, c_2^i), \ldots, (b_{M_k}^i, c_{M_k}^i) \right\} \tag{3}$$

where $b_j^i$ and $c_j^i$ denote the bounding box of *jth* object in $x_i$ and the class, respectively.

Algorithm: Optimize the loss function $L$ of classification $L_{cls}$ and bounding-box regression $L_{box-reg}$:

$$L = L_{cls} + L_{box-reg} \tag{4}$$

Formally, $L_{box}$ is based on the sum of squared errors (SSE) loss function, and $L_{cls}$ is based on the cross-entropy loss function. The loss function is optimized by training the neural network after a specific amount of epochs.

Prediction: For $x_{test}^i$, the prediction result is $y_{pred}^i$,

$$y_{pred}^i = \left\{ (b_{pred_1}^i, c_{pred_1}^i, p_{pred_1}^i), (b_{pred_2}^i, c_{pred_2}^i, p_{pred_2}^i), \ldots \right\} \tag{5}$$

where $b_{pred_j}^i$, $c_{pred_j}^i$, $p_{pred_j}^i$ are results of the bounding box, object class, and reliability. For filtering the object detection results, we use a predefined threshold that compares the reliability.

Evaluation metric: The primary metric used to evaluate the object detection algorithms' performance is the mean average precision ($mAP$). This metric considers the prediction of correct category labels and accuracy location. There are two main performance evaluation criteria: precision ($P$) and recall ($Recall$). The statistic of true positives ($TP$), false positives ($FP$), true negatives ($TN$), and false negatives ($FN$) are needed to measure the $P$ and $Recall$ values of the network model in the testing phase. The intersection-over-union ($IoU$) is a critical concept to determine whether the test results are correct or not. $TP$, $FP$, $TN$, and $FN$ values depend on the $IoU$ threshold. The formula of $IoU$ is defined in Equation (6).

$$IoU(b_{pred}, b_{lb}) = \frac{Area(b_{pred} \cap b_{lb})}{Area(b_{pred} \cup b_{lb})} \tag{6}$$

The $P$ and $Recall$ of each category of one image can be calculated as follows:

$$P_{C_{ij}} = \frac{TP_{C_{ij}}}{TP_{C_{ij}} + FP_{C_{ij}}} \tag{7}$$

$$Recall_{C_{ij}} = \frac{TP_{C_{ij}}}{TP_{C_{ij}} + FN_{C_{ij}}} \tag{8}$$

where $P_{C_{ij}}$ and $Recall_{C_{ij}}$ represent *Precision* and *Recall* of category $C_{ij}$ in the $jth$ image, respectively. The average precision ($AP$) of the category $C_i$ can be calculated:

$$AP_{C_i} = \frac{1}{m} \sum_{j=1}^{m} P_{C_{ij}} \tag{9}$$

The dataset has multiple categories, the $mAP$ of the entire category can be calculated as follows:

$$mAP = \frac{1}{n} \sum_{i=1}^{n} AP_{C_{ij}} \tag{10}$$

There are also many other criteria, but in this work's scope, the performance evaluation is measured mainly by the $mAP$ metric.

### 2.2.2. Backbone: Feature Extractor Based on MobileNetV3 and Feature Pyramid Networks

MobileNetV3: It is important to emphasize the integration of the MobileNetV3 model into the faster R-CNN architecture by its suitability for optic inspection systems [29]. Most hardware of the inspection systems are low resource use cases, therefore, mobile-friendly models should be applied to reduce latency. MobileNetV3 backbone plays the role of a feature extractor in object detectors. MobileNetV1 [30] proposed depth-wise separable convolution to reduce the number of parameters to improve computation efficiency, and MobileNetV2 [31] introduced the inverted residual block to expand to a higher-dimensional feature space internally to make more efficient layer structures. MobileNetV3 inherited advances of V1 and V2; it deployed the Squeeze-and-Excitation [32] in the inverted residual bottleneck and flexibly used the h-swish nonlinearity to significantly improve the accuracy of neural networks.

The inverted residual is the main building block of the MobileNetV3 network. The block follows a narrow–wide–narrow approach by input–output channels. As an example in Figure 14, the input channel is 24, the space expansion channel is 72, and the output channel is 40. The inverted residual block uses a combination of the expand convolution, the depth-wise convolution, the squeeze-excitation block, and the projection convolution, as in Figure 14.

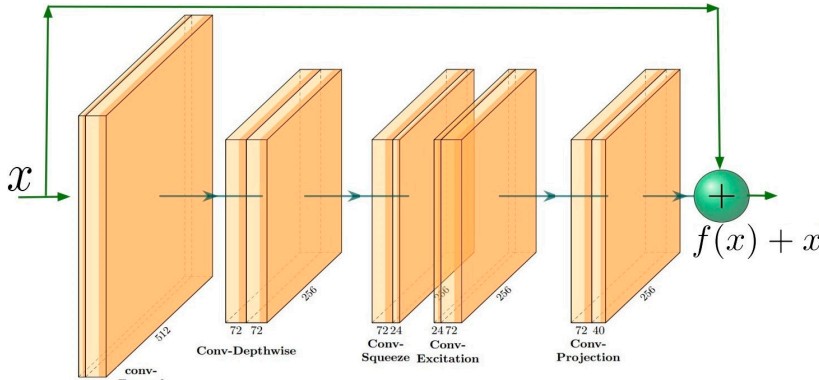

**Figure 14.** This is an instant of the inverted-residual block in MobileNetV3 architecture. First is a convolutional expand layer that widens channels from 24 to 72. Second is the convolutional depth-wise layer for better efficiency than traditional convolution. Its input and output channels are equal to 72, and the striking attribute of convolution halves the resolution. Next is the squeeze and excitation module to improve the power of features in the network. The final convolutional projection layer presents features in the lower dimension space, from 72 to 40.

FPN [33]: The object detection field has many more innovative algorithms, but current image data have become much more challenging, for instance, small object detection issues with only a few pixels. It is hard to extract the information about small objects in feature maps. FPN proposes a method to improve small object detection performance. It is an essential component that exploits the features of small objects on different levels of feature maps. FPN is an extended idea of pyramidal feature hierarchy that its architecture is a combination between top–down pathway, bottom–up pathway, and lateral connections.

As in Figure 15, the backbone architecture combines the MobilenetV3 and FPN. It is to extract multi-scale feature maps from the input image. The input is fed to MobilenetV3, which has 15 inverted residual blocks. The output is the multi-scale feature maps $\{C_1, C_2, C_3, C_4, C_5\}$, which are the input for FPN. FPN convolute and upsample $C_i$ to output the better quality multi-scale feature maps $\{P_1, P_2, P_3, P_4, P_5\}$.

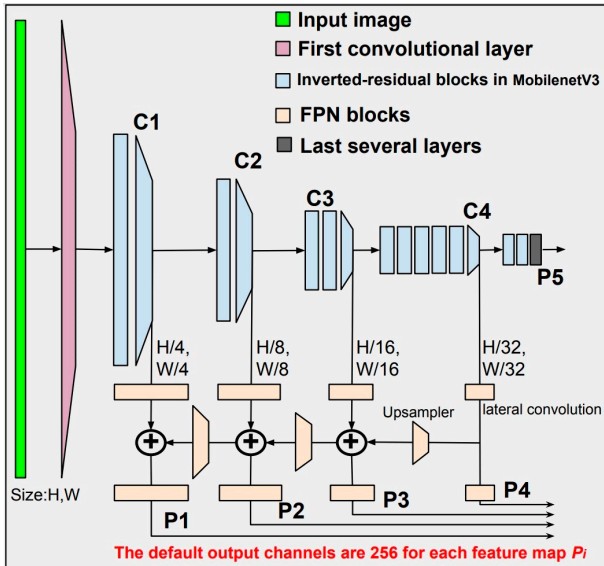

**Figure 15.** Backbone: the feature pyramid network and MobileNetV3 backbone together. The input is the image of size $H, W$. Firstly, The MobileNetV3 extracts the image to many multi-scale feature maps $\{C_1, C_2, C_3, C_4, C_5\}$. Secondly, the multi-scale output of MobileNetV3 is the input for FPN. The final result is the feature maps at multiple levels $\{P_1, P_2, P_3, P_4, P_5\}$.

The output channels of FPN present the multi-scale feature maps $\{P_1, P_2, P_3, P_4, P_5\}$. This hyper-parameter is vital to guarantee the quality of feature maps. Its default value is 256 for the large benchmark dataset. With the customized dataset, we will fine-tune the number of the output channels to obtain better performance. The results are shown in Section 3.2.

The output feature maps from FPN are $\{P_1, P_2, P_3, P_4, P_5\}$, which are also the input to the feature pyramid network and the ROI-Head. In the next section, we will describe these two blocks in detail.

### 2.2.3. RPN: Region Proposal Network

Detecting the position of objects is one of the two main tasks in object detection. The theoretical basis for initializing the temporary object position remains more challenging. In classical computer vision, selective search [34], multiscale combinatorial grouping [35], and CPMC [36] apply a strategy based on grouping super-pixels. EdgeBoxes [37] and objectness in Windows [38] use the window scoring technique. In deep learning-based computer vision, Shaoqing et al. [22] propose region proposal networks to create the box anchors to filter the potential positions. Anchor boxes are defined by two parameters: the wide range of scales and the aspect ratios.

RPN initialized a set of anchor boxes on each image or feature map by two hyper-parameters: scales and aspect ratio. They have a large impact on the final accuracy. Hence, we try to exploit them for optimal results. Figure 16 illustrates the creation of anchors on an image. The left image is an original, consisting of the red defect labels. The right image initializes a set of anchor boxes with scales $\{32^2, 64^2, 128^2, 256^2, 512^2\}$ and a range of aspect ratios $\{1 : 2, 1 : 1, 2 : 1\}$. Anchors are white, black, yellow, green, and blue rectangular boxes in the right image.

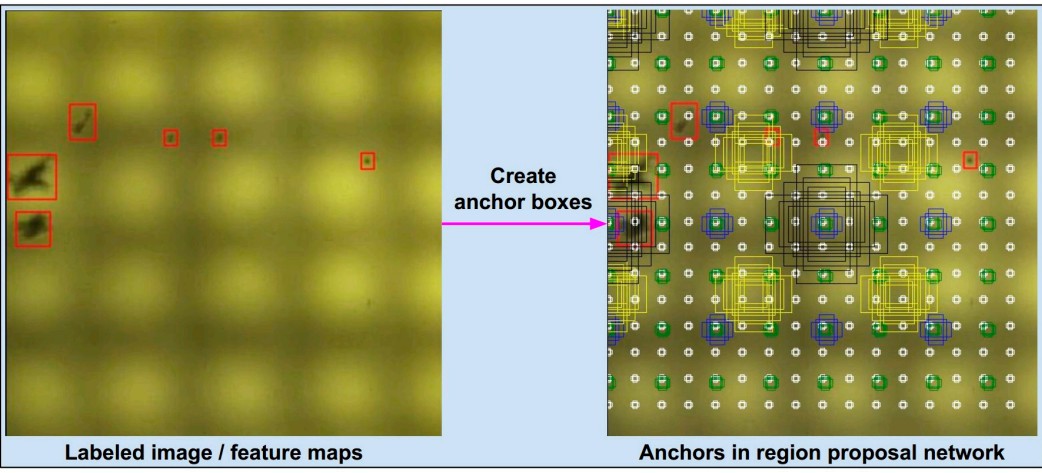

**Figure 16.** Illustrate how to create the box anchors in the region proposal network. Left is the image containing some red ground-truth boxes. RPN generates the reference boxes called "anchors" to map to ground-truth boxes. The multi-scale anchors are generated on the right image at various positions. They are the rectangular boxes marked with white, black, yellow, green, and blue colors.

The number of anchors generated is copious. RPN now tries to find anchors similar to the ground boxes (labels). The metric that determines whether an anchor is similar to the ground boxes is the *IoU* calculation. The pre-defined *IoU* thresholds are set to label the anchors as foreground, background, or ignored. If *IoU* is larger than the first threshold (typically 0.7), the anchor is assigned to one of the ground-truth boxes and labeled as foreground ('1'). If *IoU* is smaller than the second threshold (typically 0.3), the anchor is either labeled as a background ('0') or otherwise ignored ('−1').

In practice, the majority of anchors are background (the label is "0"), and so it is difficult to learn the foreground anchors due to the label imbalance. To solve the imbalance issue, the target number of foreground boxes $N$ and the target number of background $M$ are pre-defined.

At this time, we have the labeled anchor set and a target set, as shown in Figure 17. The RPN should learn to find rules to recognize the exact locations and shapes of ground-truth boxes. This issue is known as bounding-box regression, which will be presented in the next section.

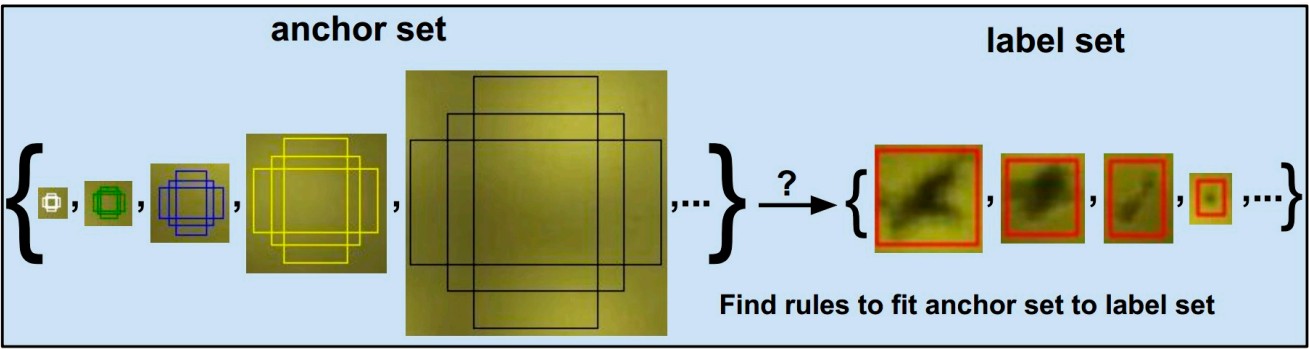

**Figure 17.** Box regression transforms the proposal anchor set to ground-truth label set.

The role of RPN is to propose the potential regions that contain the defects. To achieve this task, training the RPN is to regress the anchor boxes to the defect regions and classify anchors as the labels "1" or "0". Because of the large number of anchors, we only choose some of the quality anchors called *proposals* for the next sub-network. The loss function of RPN includes the $L_1$ [20] loss function for bounding-box regression and binary cross-entropy loss function for classifying the anchor as ground-truth or background.

$$L_{RPN} = L_{box-reg-RPN} + L_{binary-cls} \tag{11}$$

The following section details the bounding box regression and its loss function.

### 2.2.4. Bounding Box Regression

Bounding box regression will find some rules to scale-invariant transform a bounding box (anchor) to another bounding box (ground-truth/defect). The best idea is to consider the relationship between the center coordinates, where their width–height dimensions are significant. This section describes the formula between the ground-truth box and anchor. Figure 18 illustrates the parameters involved in transforming an anchor (the blue dotted-line rectangle) into a ground-truth box (the green rectangle) during the training phase, with the parameters are calculated using Equation (12).

$$\begin{aligned} \delta_x &= (b_x - a_x)/a_w, & \delta_y &= (b_y - a_y)/a_h \\ \delta_w &= \log(b_w/a_w), & \delta_h &= \log(b_h/a_h) \end{aligned} \tag{12}$$

where "$a$" and "$b$" denote anchor box and ground-truth box, respectively. Each one is represented by a 4-tuple in the form of $(x, y, w, h)$, where $(x, y)$ is the center coordinate and $(w, h)$ is the width and height dimension. The regressor $f$ aims to predict the transformation $\delta$ from the anchor $a$ to the target ground-truth box $b$, represented as follows:

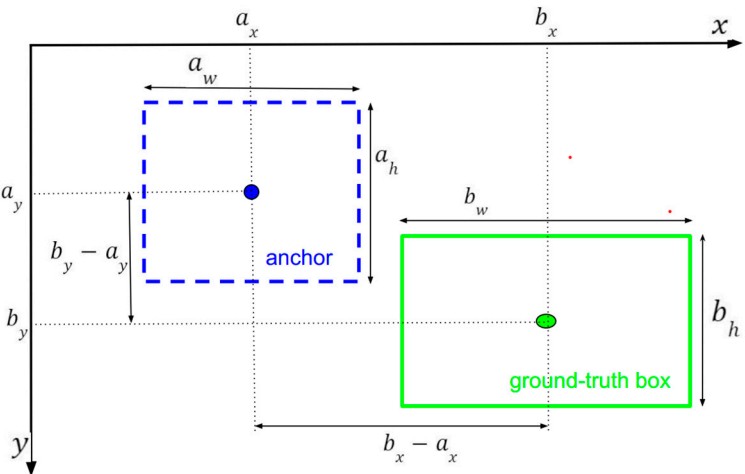

**Figure 18.** Illustration of transformation $\delta$ from the anchor $a$ to the ground-truth box $b$. The formula is in Equation (12).

The image feature, denoted as $x$, is used as the input for the regressor $f$. Consiquently, the output is a prediction represented by $\hat{\delta} = f(x)$. The training process will minimize the bounding-box loss function:

$$L(\hat{\delta}, \delta) = \sum_{p \in \{x,y,w,h\}} L_1^{smooth}(\hat{\delta}_p - \delta_p),  \tag{13}$$

where the function $L_1^{smooth}(.)$ is the robust $L_1$ loss defined in Equation (14).

$$L_1^{smooth}(t) = \begin{cases} 0.5t^2 \, if \, |t| < 1 \\ |t| - 0.5 \, otherwise \end{cases}  \tag{14}$$

To calculate the final prediction box coordinates, the regressed anchor is inferred based on the inverse transformation of Equation (15) as follows:

$$\begin{array}{ll} a_x^{pred} = \hat{\delta}_x a_w + a_x, & a_y^{pred} = \hat{\delta}_y a_h + a_y \\ a_w^{pred} = a_w \exp(\hat{\delta}_w), & a_h^{pred} = a_h \exp(\hat{\delta}_h) \end{array}  \tag{15}$$

The final summary is as follows: the bounding-box regressor $f$ is a neural network with the input $T$, which are the image or feature maps, and the label is $\delta$. A prediction is $\hat{\delta} = f(T)$. The training process will optimize the loss function $L_1^{smooth}(\delta - \hat{\delta})$. With the formulas $\delta$, $L_1^{smooth}$ and $\hat{\delta}$ in Equations (12) and (14), and formula $\hat{\delta} = f(T)$, respectively.

### 2.2.5. ROI-Head

ROI-Head converts the selected proposals on each feature map into a small fixed window (usually $7 \times 7$ pixels), and next is fed to the linear neural network to regress the bounding boxes and classify defects.

The inputs of ROI-Head are: The feature maps $\{P2, P3, P4, P5\}$ from the backbone block; the proposal boxes from the RPN block; and the label of defects. The ratio of foreground and background boxes will be customized to accelerate the training. The proposals with higher *IoU* than the threshold are counted as foreground and the others as background. This step will choose the best k proposals based on the *IoU* metric.

Before entering the ROI process, the top-k proposals are assigned to each level $P_i$ of the appropriate feature maps based on the formula in Equation (16).

$$L_{P_i} = floor(k_0 + \log_2(\frac{\sqrt{w * h}}{\text{canonical\_box\_size}}))  \tag{16}$$

where $k_0$ is the reference value, which is generally set to 4; $w$ and $h$ are the the width and height of the ROI area, respectively; *and* canonical_box _size is the canonical box size in pixels, set to 224, corresponding to the size of the pre-training image of the ImageNet dataset.

The ROIAlignV2 [37] process crops the rectangular regions on the feature maps specified by the proposal boxes. The linear neural network feeds the results of ROI to regress the proposals to ground boxes and classify the defect type.

When training the ROI-Head network, the loss function sums up the cost of classification $L_{defect-cls}$ and bounding-box regression $L_{box-reg-ROI}$, as in Equation (17).

$$L_{\text{detector}} = L_{defect-cls} + L_{box-reg-ROI} \tag{17}$$

where $L_{defect-cls}$ is the defect classification loss function that computes the cross-entropy loss; and $L_{box-reg-ROI}$ is the smooth $L_1$ loss as in Equation (13).

### 2.2.6. End-to-End Learning

The RPN block needs the cost to classify anchors as background or foreground (binary cross-entropy) and find proposals for the candidate locations of defects (bounding-box regression). Meanwhile, the ROI-Head network also incurs a cost to classify the defect type (cross-entropy) and locate the defects' position (bounding-box regression). Therefore, the network can be trained in an end-to-end manner using the multi-task loss function as follows:

$$L = L_{RPN} + L_{detector} \tag{18}$$

where $L_{RPN}$ and $L_{detector}$ are based on the formulae in Equations (7) and (13), respectively.

### 3. Results

#### 3.1. Experimental Setting

Defects are labeled and converted to COCO format to be compatible with object detection models. All images are resized to 1333 px for long edge and 800 px for short edge. We split the dataset into the training and test sets by a ratio of 80:20. In the first step, we train and test on some of the standard defect detection architectures such as two-stage object detectors (Faster-RCNN) and one-stage object detectors (Retina, FCOS). All models are implemented using PyTorch Vision's default configuration [39]. Table 6 displays the experimental outcomes obtained from training Faster R-CNN-based models with ResNet50, MobileNetV3-large, and MobileNetV3-small backbones, as well as the RetinaNet model, and the FCOS model, using the initial dataset. In the second step, for increment accuracy, we fine-tune the hyper-parameter in RPN, while for computational efficiency, we reduce the output channel of FPN in the backbone. The final result is presented in the following section.

**Table 6.** Comparison of defect detection between each architecture trained on the ski goggles defect dataset without any hyper-parameters adaptation.

| Architecture | BACKBONE | IoU Metric | | | SPEED (S/IT) | |
|---|---|---|---|---|---|---|
| | | AP | AP$_{50}$ | AP$_{75}$ | TRAIN | TEST |
| | RESNET50 | 56.3 | 78.5 | 63.3 | 0.528 | 0.126 |
| Faster-RCNN | MOBILE-LARGE | 41.3 | 72.8 | 38.1 | 0.127 | 0.059 |
| | MOBILE-SMALL | 10.0 | 25.1 | 08.4 | 0.086 | 0.045 |
| FCOS | RESNET50 | 59.6 | 78.6 | 64.0 | 0.352 | 0.126 |
| RetinaNet | RESNET50 | 10.2 | 25.2 | 05.9 | 0.331 | 0.140 |

The models are trained with 2 GPUs with a batch size of 8 for 26 epochs using SGD optimizer. The learning rate is initialized to 0.02 and learning ratio step at 16 and 22. Computer configuration is a CPU AMD Ryzen5 3600X, 64G RAM, 2 GPUs Gigabyte 2060 6G.

The method evaluates detection results based on the standard COCO-style average precision measured at *IoU* thresholds ranging from 0.5 to 0.75.

### 3.2. Defect Detection Results

To have a defect detection result baseline for the defect dataset on the surface of ski goggles lenses, we train and test different architectures and backbones. Parameters of Faster-RCNN-ResNet50, Faster-RCNN-Mobile-large, Faster-RCNN-Mobile-small, FCOS-Resnet50, and Retina-Resnet50 models are 41.1 M, 18.9 M, 15.8 M, 31.85 M, and 32.05 M, respectively. Table 6 shows the linear result of the larger architecture (more parameter) having better accuracy, but slower computational efficiency (speed, s/it). The balance between accuracy and computational efficiency is an issue in automatic optical inspection as hardware characteristics are compact. We recognize that the Faster-RCNN-Mobile-large model has gained a balanced result in terms of accuracy and computational efficiency. From this, we decide to fine-tune the Faster R-CNN with backbone Mobile-large to achieve a better result.

Faster R-CNN has proven to be a state-of-the-art object detector with high accuracy and flexible modular ability. Therefore, it can be integrated into some sub-network to improve performance. We implement the Faster R-CNN-based detector that uses an FPN-style backbone that extracts features from different convolutions of the MobileNetV3 model. The advance of MobileNetV3 block helps to improve speed; alternatively, FPN presents the invariant of feature maps, leading to an improvement in the small defect detection. However, Faster R-CNN has a drawback due to the complicated computation in creating anchor boxes. Its hyper-parameters in RPN are often sensitive to the final detection performance. The above disputation leads to fine-tuning Faster R-CNN to archive high performance.

First, we fine-tune the output channel of FPN to improve the network's speed. All feature maps extracted from the MobileNetV3 network have their output projected down to the number of channels by the FPN block. The default number of the output channel is 256. This parameter is finetuned within the value set $\{256, 128, 96, 64\}$ to obtain the best possible performance.

Second, we fine-tune the *anchor scale factor* in RPN to improve the accuracy. The *anchor scales* affect the handling of the bounding boxes of different sizes. Its invalid value setting causes the imbalance between negative and positive samples in training. The default value of *anchor scales* in the Faster R-CNN model is $\{32^2, 64^2, 128^2, 256^2, 512^2\}$. We augment two values: $\{16^2, 32^2, 64^2, 128^2, 256^2\}$ and $\{8^2, 16^2, 32^2, 64^2, 128^2\}$ in the RPN block.

Table 7 shows the aggregate results of fine-tuning the *output channel number* in FPN and the *anchor scales* in RPN. When the output channel of FPN decreases, the training and testing speed improves, while accuracy slightly decreases. Observing the efficiency of anchor scales, configuration $\{16^2, 32^2, 64^2, 128^2, 256^2\}$ achieved better results than the other two configurations, $\{32^2, 64^2, 128^2, 256^2, 512^2\}$ and $\{8^2, 16^2, 32^2, 64^2, 128^2\}$, with the same channel as FPN. With the channel reduction in FPN from 256 to 128, and the replacement of the anchor scale $\{32^2, 64^2, 128^2, 256^2, 512^2\}$ by $\{16^2, 32^2, 64^2, 128^2, 256^2\}$, the accuracy is 55.0 $mAP$, which is close to the best accuracy, while the $AP_s$ metric achieved the best accuracy with 47.0. From this result, we choose the optimistic parameter set with the output channel equal to 128 and the *anchor scales* equal to $\{16^2, 32^2, 64^2, 128^2, 256^2\}$ for balance in computational efficiency and accuracy.

As mentioned in Section 2.1.3, the combined dataset (CoDS) arises from the fusion of initial (InDS) and synthetic datasets. Comparing these datasets is crucial to illustrate the efficacy in addressing a few data and imbalances. Table 8 depicts the evaluation outcomes for both datasets when training the Faster R-CNN model with optimal parameters.

**Table 7.** Defect detection results from fine-tuning the output channel of FPN and the anchor scale in RPN.

| FPN | RPN | IOU METRIC | | SPEED (S/IT) | |
|---|---|---|---|---|---|
| Out Channel | Anchor Scales | MAP | APS | TRAIN | TEST |
| 256 | $\{8^2, 16^2, 32^2, 64^2, 128^2\}$ | 55.3 | 46.4 | 0.4864 | 0.1161 |
| 256 | $\{16^2, 32^2, 64^2, 128^2, 256^2\}$ | 49.2 | 42.8 | 0.4867 | 0.1179 |
| 128 | $\{8^2, 16^2, 32^2, 64^2, 128^2\}$ | 53.6 | 45.4 | 0.3040 | 0.1133 |
| 128 | $\{16^2, 32^2, 64^2, 128^2, 256^2\}$ | 55.0 | 47.0 | 0.3080 | 0.1074 |
| 96 | $\{8^2, 16^2, 32^2, 64^2, 128^2\}$ | 46.7 | 39.6 | 0.2857 | 0.1094 |
| 96 | $\{16^2, 32^2, 64^2, 128^2, 256^2\}$ | 51.8 | 42.7 | 0.2860 | 0.1046 |
| 64 | $\{8^2, 16^2, 32^2, 64^2, 128^2\}$ | 47.6 | 38.3 | 0.2517 | 0.0993 |
| 64 | $\{16^2, 32^2, 64^2, 128^2, 256^2\}$ | 51.4 | 46.0 | 0.2520 | 0.0968 |

**Table 8.** The comparative assessment of the initial dataset and the combined dataset using COCO metrics.

| Model | COCO Metric | DATASET | |
|---|---|---|---|
| | | INITIAL DS (InDS) | COMBINED DS (CoDS) |
| Faster R-CNN with the | AP | 51.4 | 55.1 |
| MobileV3 Backbone | $AP_{50}$ | 71.5 | 75.8 |
| | $AP_{75}$ | 40.7 | 47.4 |
| The output channel number of | $AP_S$ | 46.0 | 48.2 |
| FPN is 64 | $AP_m$ | 47.3 | 50.3 |
| | $AP_l$ | 59.1 | 63.4 |
| The anchor scales of RPN | $AR_1$ | 31.3 | 32.6 |
| $\{16^2, 32^2, 64^2, 128^2, 256^2\}$ | $AR_{10}$ | 50.9 | 53.8 |
| | $AR_{100}$ | 55.6 | 59.4 |
| | $AR_S$ | 45.2 | 49.7 |
| | $AR_m$ | 61.6 | 54.6 |
| | $AR_l$ | 60.4 | 64.9 |

Figure 19 shows some results on the test set of the lens defect dataset. The red rectangular boxes mark defects; the above label is the defect category.

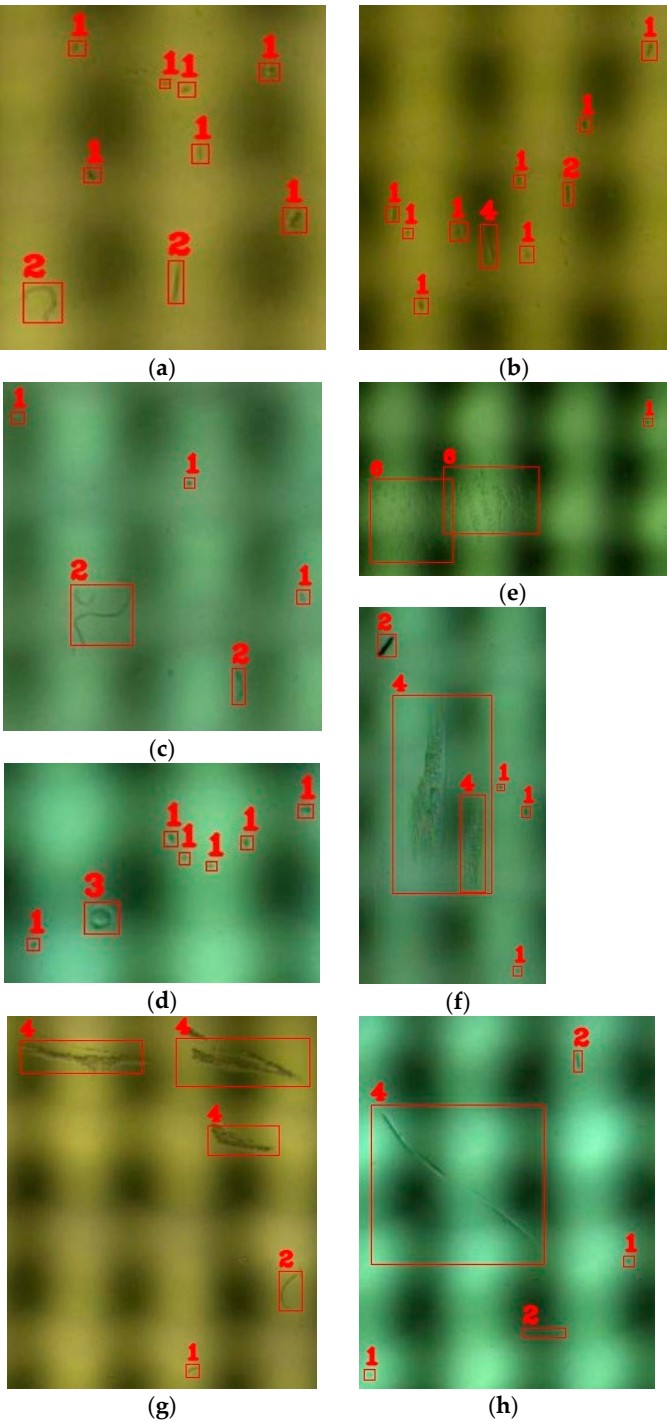

**Figure 19.** Selected examples of defect detection results. Defects are marked by the red rectangular boxes and the label above is the defect category. (**a**–**c**) display a variety of defect types, including dust-spot and dust-line. (**d**) showcases the spotlight defect type, while (**e**) highlights the stain defect type. (**f**–**h**) and h feature the scratch defect type.

## 4. Discussion

The data presented in Tables 6 and 7 have been computed using the detection evaluation metrics employed in the COCO detection challenge. A comparative analysis of object detection models, such as Faster R-CNN, FCOS, and RetinaNet, was conducted based on the data provided in Table 6. The Faster R-CNN model serves as a baseline for comparison due to its widespread use in object detection tasks. In terms of precision, FCOS

outperforms Faster R-CNN with a 44.31% improvement in AP, a 7.97% enhancement in AP50, and a 67.98% increase in AP75. However, FCOS exhibits slower training and testing speeds, with a 177.17% reduction in training and a 113.56% decrement in testing relative to Faster R-CNN. Conversely, RetinaNet shows a significantly lower precision performance, with a 75.30% reduction in AP, a 65.35% decrease in AP50, and an 84.51% decline in AP75, while also demonstrating slower training and testing speeds (160.63% and 137.29% slower, respectively), in comparison to Faster R-CNN.

In the context of an optical inspection system, it is crucial to prioritize solutions that demand minimal hardware resources while maintaining satisfactory performance levels. A comprehensive analysis of various object detection models, including Faster R-CNN, FCOS, and RetinaNet, reveals that Faster R-CNN emerges as the most suitable candidate for deployment. Despite FCOS exhibiting superior overall precision, it significantly necessitates more hardware resources owing to its slower training and testing speeds, with a 177.17% reduction in training and a 113.56% decrement in testing compared to Faster R-CNN. In contrast, Faster R-CNN strikes an optimal balance between performance and resource efficiency, featuring notable precision performance and faster training and testing speeds in comparison to FCOS and RetinaNet. Consequently, Faster R-CNN is the preferable choice for deployment when considering hardware resource constraints.

Additionally, in Table 6, we dive into the performance analysis of some backbones of the Faster R-CNN model. Comparing speeds between ResNet50, MobileNet-Large, and MobileNet-Small as Faster R-CNN backbones showcases the trade-offs between performance and resource efficiency. Although the ResNet50 backbone outperforms the MobileNet-Large backbone in terms of AP, AP50, and AP75, it is notably slower in both training and testing phases, with training taking 315.75% longer and testing experiencing a 113.56% increase in duration. In contrast, MobileNet-Small provides the quickest training and testing speeds among the backbones, at 32.28% and 23.73% faster speeds than MobileNet-Large. However, this speed advantage comes at the cost of significantly reduced performance metrics. MobileNet-Large balances performance and speed, maintaining competitive performance metrics while achieving relatively faster training and testing speeds than ResNet50.

The optimal model for this context combines the strengths of Faster R-CNN, FPN, and MobileNetV3. The next step in optimizing this model is to fine-tune two parameters: the *output channel number* in FPN and the *anchor scales* in RPN. As shown in Table 7, the best-performing use-case has an FPN *output channel* of 256 and RPN *anchor scales* of $\{8^2,$ $16^2, 32^2, 64^2, 128^2\}$, with a *mAP* score that is 12.28% higher than the second-best use-case. Furthermore, the best-performing use-case has an $AP_S$ score of 46.4, which is 8.62% higher than the second-best use-case. In cases where speed is prioritized, the fastest use-case is the one with an FPN *output channel* of 64 and RPN *anchor scales* of $\{16^2, 32^2, 64^2, 128^2, 256^2\}$. Compared to the best-performing use-case, this use-case is 48.78% faster in training and 16.28% faster in testing.

Table 8 compares two InDS and CoDS datasets utilizing various COCO metrics. Dataset CoDS exhibits superior performance in the majority of metrics when compared to dataset InDS. Notably, CoDS surpasses InDS in average precision (*AP*) with a 7.2% increase, as well as in size-based subcategories ($AP_s$, $AP_m$, *and* $AP_l$), with enhancements ranging from 4.8% to 7.3%. Regarding average recall (AR), dataset CoDS exceeds InDS in most categories, except for the medium object size category ($AR_m$), where InDS outperforms CoDS by 11.4%. In summary, the combined dataset demonstrates improved performance relative to the initial dataset.

Imbalanced datasets and scarce data are common challenges when training deep learning models with real-world data collection. CoDS performs better than InDS, thereby illustrating that generating additional synthetic data is an effective approach to addressing these challenges.

Despite the valuable contributions of this study, certain limitations should be acknowledged: the number of labels in the defect dataset is relatively small, which may

impact the generalizability of the findings; the approach to addressing unbalanced and rare datasets is not extensively explored; the study does not employ newer defect detection methods within deep learning to analyze the dataset; and a limited range of metrics is considered for a detailed evaluation of defects, suggesting that additional performance measures could provide further insight. Future research may address these limitations by expanding the dataset, exploring more comprehensive solutions for unbalanced and rare data, incorporating cutting-edge defect detection techniques, and utilizing a wider array of evaluation metrics.

## 5. Conclusions

This paper solved the defect detection problem on the surface of ski goggles lenses based on the deep learning approach. The work has achieved the design of the image capture system that has five cameras cover the entire curved surface of the lenses, which enables it to capture images automatically from all angles at the same time. This work also presents the development of a surface defect detection dataset for ski goggles lenses, contributing to the diversification of surface data sources in the deep learning-based defect detection field. The defect detection result achieved excellent performance by fine-tuning the reasonable hyper-parameters of the Faster-RCNN modular architecture by replacing the ResNet backbone with MobileNetV3 and FPN to better extract feature maps, by reducing the number of the output channel of FPN to increase the computational performance, and by adjusting the anchor scale factor hyper-parameter in RPN, leading to better accuracy. This work is helpful for automatic optical inspection systems because of its limited hardware resource. The experimental results have reinforced the hypothesis for correctly choosing the Faster-RCNN defect detection architecture and fine-tuning the hyper-parameters. In the future, we will improve the dataset and make it publicly available to the research community.

**Author Contributions:** D.-T.D. developed the hardware system and software coding, and wrote the original draft. J.-W.W. guided the research direction and edited the paper. All authors discussed the results and contributed to the final manuscript. All authors have read and agreed to the published version of the manuscript.

**Funding:** This research was funded in part by MOST 109-2218-E-992-002 from the Ministry of Science and Technology, Taiwan.

**Data Availability Statement:** The datasets used in this study can be accessed through the following link: https://github.com/ddthuan/goggles (accessed on 30 January 2023).

**Acknowledgments:** The authors would like to thank Foresight Optical Co., Ltd., Taiwan, for their joint development of the inspection system.

**Conflicts of Interest:** The authors declare no conflict of interest. This article is authorized for the first author to continue with all his related study works.

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
