# Peer review of "Developing a Deep Learning-Based Defect Detection System for Ski Goggles Lenses"

_axioms, doi:10.3390/axioms12040386_

Round 1

Reviewer 1 Report

One of the problems in the manufacturing process of ski goggles is defects occurring on the surface of the lenses. Therefore, in order to improve product quality, goggle manufacturers are forced to implement effective quality control systems. To solve this problem, the authors proposed a comprehensive defect detection system that includes key stages such as an image acquisition module and a defect detection module. In this context, the original image acquisition system deserves attention, which may be of interest to other researchers working on similar problems. To build the defect detection module, the authors applied the MobileNetV3 backbone used in a Feature Pyramid Network along with the Faster-RCNN detector. In this area, the authors have clearly and comprehensively explained the architecture of the system and the methodology used. The accuracy and computational efficiency achieved by the authors of the proposed system allows it to be used to solve the problem of detecting lens defects.

The problem raised by the authors is an important one, and the obtained results make a valuable contribution to the research on automatic optical inspection systems. The manuscript is well written and it should be of great interest to the readers. I recommend accepting the manuscript for publication in its present form.

Author Response

Dear Reviewer,

First and foremost, we would like to thank you so much for carefully reviewing this manuscript. Without your constructive and valuable comments, suggestions, this manuscript cannot be improved. Therefore, we greatly appreciate your time and support. We believe that our work will benefit the readers and contribute to the advancement of knowledge in the field.

Thank you once again for your time and consideration.

Best regards!

Dinh-Thuan Dang

Reviewer 2 Report

Paper is very good and a small suggestion is please avoid using the second person and third person words such as "we", "our" etc,.

Author Response

Dear Reviewer,

Thank you for the positive feedback on our manuscript. We appreciate your kind words regarding our paper and your valuable suggestion to refrain from using second and third person pronouns such as "we" and "our." We recognize the importance of adopting an objective tone in academic writing.

Following your advice, the manuscript has been updated to eliminate these pronouns to conform to the academic style.

Once again, thank you so much for your time reviewing this manuscript and your valuable suggestions.

Yours faithfully,

Dinh-Thuan Dang

Reviewer 3 Report

The paper presents a design of ski goggles inspection system, where the authors introduce a five-camera setup and software for defect detection on the lenses. As expected for a modern vision system, some deep learning-based alternatives were selected and tested and the most efficient was selected as a possible solution. For their investigation authors collected a dataset of examples. The dataset was unbalanced, however, no detailed analysis of the effect of the unbalanced dataset to detection results detection is provided in the paper.

The paper includes a big portion of the text where basic fundamentals are explained. This increases the size of the paper, however, do not add scientific contribution which is very low in this paper.  The presented system is fine-tuned, uses modern vision tools, works with limited precision. However, readers probably expect to get new original ideas, on how to make such systems more accurate, more automated, robust, etc. Such part is missing or expressed in a very limited way in this paper. 

Author Response

Dear Reviewer,

We sincerely appreciate the time and effort you have taken to review our manuscript. We value your feedback and would like to address the concerns you raised in your comments.

  • Unbalanced dataset and its effect on detection results
  • Basic fundamentals and scientific contribution.

Once again, we would like to thank you for your constructive feedback. We believe that your valuable insights will help us improve the quality and impact of our manuscript. We look forward to incorporating these suggestions in our revised submission.

Sincerely,

Dinh-Thuan Dang

Reviewer 4 Report

The authors present a novel idea of detecting defects in ski googles. There some deficiencies that need to be accounted for before accepting the manuscript.

ABSTRACT

What is mAP?

Related Work

Figure 3. The resolution needs to be enhanced

Section3 2.1-2.2: there is some redundancy in these two sections. There is no need to go in details for the basic of computer vision. The authors need to focus more in the studies that worked in the same topic.

The authors MUST add a table that contain the number of images for each defect category.

How did the authors deal with the insufficient number of images to build a deep learning model? Did you implement augmentation?

In Figure 5, how was the ROI extracted in the image acquisition section?

How can you decide the points of defects can’t be affected by surface reflection?

The authors presented a well written and detailed methods for the experimental design and methods of analysis

The authors must compare their results to previous work even with different technologies. This is a major drawback in this study.

Conclusions: the authors need to add some numbers.

Author Response

Dear Reviewer,

First and foremost, we would like to thank you so much for carefully reviewing this manuscript. Without your constructive and valuable comments, suggestions, this manuscript cannot be improved. We appreciate the time and effort you have devoted to reviewing our work. We have carefully considered your comments and would like to address each of them: some redundancy, images for each defect category, the insufficient number of images, extract ROI, surface reflection.

We are grateful for your insightful comments and suggestions, which will undoubtedly contribute to improving the quality and impact of our manuscript. We look forward to incorporating these changes in our revised submission.

Sincerely,

Dinh-Thuan Dang

Reviewer 5 Report

The article prepared by the authors seems to be a very interesting, even innovative research and application activity. Currently, machine and deep learning is a powerful development tool that gives new expectations to the industry for the future. For my part, the article is acceptable but requires a lot remarks to improve clarity for the reader:

Line 26: The authors used the term "mAP" which they did not declare earlier in the text. Please explain what it means.

Line 33. Correct the language style in the sentence.

Line 38. Correct the language style in the sentence. Remove "its" in this sentence.

Line 40. Correct the language style in the sentence.

There is no continuity in reading the text between the sentence on line 46 and the next sentence on line 47. Correct it.

I think that Figures 1 and 2 should be in the Material and Methods section. This also involves the transfer of the description. Correct it.

The article requires the preparation of the correct structure for individual sections and the provision of appropriate headings. Especially since a lot of methods have been presented. This involves correcting the sentences in the introduction where the individual stages (sections) of the article have been listed. Correct it.

Line 75-88. This is a description that should be in the Introduction.

I think authors should remove the "Related Works" . Please move the information describing the research object to the Material section. The rest of the information relates to references that should be in the introduction.

"Figure 5 is the fundamental paradigm for our proposed approach". Is this sentence necessary? Might be worth linking the reference to "Figure 5" in the next sentence.

However, the authors need to improve the structure of the article as I mentioned earlier for clarity of the work and the journal structure must be maintained by adding a "Results and Discussion" section too.

In the summary, there are no numerical data confirming the fact of the quoted conclusions in relation to individual ARs. Correct it.

The authors emphasize that the light source is very important when performing the image acquisition process. Please explain what impact the environment has on the quality of the obtained photos in the preparation of such research activities? I mean the surroundings, the exposure parameters of the object.

Author Response

Dear Reviewer,

We express our appreciation to the reviewer for investing your valuable time in a comprehensive reading and offering constructive feedback. Without your constructive and valuable comments, suggestions, this manuscript cannot be improved.

We have carefully checked and revised our manuscript based on your suggestions. Please kindly see revised parts which are highlighted in the manuscript and also find our responses to your comments and suggestions in the below part of this letter.

We highly believe that your comments and suggestions helped us a lot in improving our manuscript.

Once again, thank you so much for your time reviewing this manuscript and your valuable comments and suggestions.

Yours faithfully,

Dinh-Thuan Dang

Round 2

Reviewer 3 Report

Authors made sufficient changes for this paper to emphasize contribution and clarify the experimental setup and justification of received results.